# Macrolides Decrease the Proinflammatory Activity of Macrolide-Resistant *Streptococcus pneumoniae*

Hisanori Domon,[a,b] Satoru Hirayama,[a] Toshihito Isono,[a] Karin Sasagawa,[a,c] Fumio Takizawa,[a,c] Tomoki Maekawa,[a,b,c] Katsunori Yanagihara,[d] Yutaka Terao[a,b]

aDivision of Microbiology and Infectious Diseases, Niigata University Graduate School of Medical and Dental Sciences, Niigata, Japan
bCenter for Advanced Oral Science, Niigata University Graduate School of Medical and Dental Sciences, Niigata, Japan
cDivision of Periodontology, Niigata University Graduate School of Medical and Dental Sciences, Niigata, Japan
dDepartment of Laboratory Medicine, Nagasaki University Graduate School of Biomedical Sciences, Nagasaki, Japan

**ABSTRACT** Over the past 2 decades, the prevalence of macrolide-resistant *Streptococcus pneumoniae* (MRSP) has increased considerably, due to widespread macrolide use. Although macrolide usage has been proposed to be associated with treatment failure in patients with pneumococcal diseases, macrolides may be clinically effective for treating these diseases, regardless of the susceptibility of the causative pneumococci to macrolides. As we previously demonstrated that macrolides downregulate the transcription of various genes in MRSP, including the gene encoding the pore-forming toxin pneumolysin, we hypothesized that macrolides affect the proinflammatory activity of MRSP. Using HEK-Blue cell lines, we found that the supernatants from macrolide-treated MRSP cultures induced decreased NF-$\kappa$B activation in cells expressing Toll-like receptor 2 and nucleotide-binding oligomerization domain 2 compared to the supernatants from untreated MRSP cells, suggesting that macrolides inhibit the release of these ligands from MRSP. Real-time PCR analysis revealed that macrolides significantly downregulated the transcription of various genes encoding peptidoglycan synthesis-, lipoteichoic acid synthesis-, and lipoprotein synthesis-related molecules in MRSP cells. The silkworm larva plasma assay demonstrated that the peptidoglycan concentrations in the supernatants from macrolide-treated MRSP cultures were significantly lower than those from untreated MRSP cultures. Triton X-114 phase separation revealed that lipoprotein expression decreased in macrolide-treated MRSP cells compared to the lipoprotein expression in untreated MRSP cells. Consequently, macrolides may decrease the expression of bacterial ligands of innate immune receptors, resulting in the decreased proinflammatory activity of MRSP.

**IMPORTANCE** To date, the clinical efficacy of macrolides in pneumococcal disease is assumed to be linked to their ability to inhibit the release of pneumolysin. However, our previous study demonstrated that oral administration of macrolides to mice intratracheally infected with macrolide-resistant *Streptococcus pneumoniae* resulted in decreased levels of pneumolysin and proinflammatory cytokines in bronchoalveolar lavage fluid samples compared to the levels in samples from untreated infected control mice, without affecting the bacterial load in the fluid. This finding suggests that additional mechanisms by which macrolides negatively regulate proinflammatory cytokine production may be involved in their efficacy *in vivo*. Furthermore, in this study, we demonstrated that macrolides downregulated the transcription of various proinflammatory-component-related genes in *S. pneumoniae*, which provides an additional explanation for the clinical benefits of macrolides.

**KEYWORDS** inflammation, lipoproteins, lipoteichoic acid, macrolide-resistant *Streptococcus pneumoniae*, macrolides, peptidoglycan

Address correspondence to Yutaka Terao, terao@dent.niigata-u.ac.jp.

The authors declare no conflict of interest.

*S*treptococcus pneumoniae, the pneumococcus, is one of the major pathogens causing community-acquired pneumonia (CAP). This organism causes severe invasive diseases, such as meningitis and bacteremia, that result in morbidity and mortality worldwide, particularly in young children, older adults, and immunocompromised patients. Although antibiotics are used as a primary treatment for pneumococcal diseases, the emergence of pneumococcal clinical isolates with resistance to penicillin, macrolides, and fluoroquinolones has considerably complicated their treatment (1). Surveillance data in the United States indicated that 32% and 36% of 590 pneumococcal isolates were resistant to penicillin and azithromycin, respectively (2). Our previous study in Japan showed that the prevalence of macrolide-resistant *S. pneumoniae* (MRSP) isolates was much higher (82%) than that in the United States (3). An increased prevalence of penicillin resistance in *S. pneumoniae* in the 1980s to 1990s shifted antibiotic treatment of pneumococcal infections to macrolides, which led to an increased prevalence of MRSP (4).

Although several case reports have highlighted macrolide treatment failure during MRSP infections like CAP and meningitis (5), azithromycin (AZM) therapy has demonstrated clinical efficacy and bacterial eradication for the treatment of CAP caused by MRSP (6, 7). Furthermore, a phase 3 clinical trial demonstrated a relatively weak relationship between the AZM MICs and clinical outcomes of *S. pneumoniae*-induced community-acquired respiratory tract infections treated with AZM monotherapy (8). Additionally, macrolide usage in patients with severe sepsis is associated with decreased mortality, even in cases where macrolide resistance has been documented (9). These findings indicate a paradox between *in vitro* macrolide resistance and *in vivo* efficacy. Therefore, macrolides may exert beneficial effects in patients with MRSP infection.

Several molecular mechanisms have been postulated to explain the beneficial effects of macrolides on MRSP pulmonary infections and a variety of noninfectious pulmonary disorders, such as diffuse panbronchiolitis and bronchiectasis (10, 11). In this regard, it has been reported that macrolides have immunomodulatory properties. Clinical and animal studies have reported that macrolides decrease the infiltration of leukocytes into inflamed lungs due to reduced levels of inflammatory cytokines and chemokines (12, 13). In addition, macrolides enhance the phagocytic ability of alveolar macrophages *in vitro* (14). In addition to their immunomodulatory properties toward host cells, macrolides affect pneumococcal cells and alter the transcription of pneumococcal genes (15). We previously demonstrated that AZM and erythromycin (ERY) downregulate the genes encoding the pore-forming toxin pneumolysin (PLY) in MRSP and decrease the pathogenicity of the bacterium (16).

In this study, we hypothesized that macrolides would affect the proinflammatory activity of MRSP. To test this hypothesis, we investigated the production of proinflammatory cytokines by human THP-1-derived macrophages stimulated with AZM- or ERY-treated MRSP. In addition, we analyzed the innate immune receptors involved in this process using the HEK-Blue cell system.

## RESULTS

**Treatment of MRSP cells with macrolides significantly reduces the proinflammatory activity of pneumococcal culture supernatants.** Our previous studies have demonstrated that sub-MICs of macrolides retard the growth of MRSP (15, 16). Therefore, to exclude macrolide-induced inhibition of pneumococcal growth, macrolide-resistant *S. pneumoniae* strain NU4471 was cultured in RPMI 1640 medium supplemented with 5 $\mu$g/mL AZM or ERY until the culture reached the stationary phase of growth (optical density at 600 nm [$OD_{600}$] of 0.55; the incubation periods of the untreated control, AZM, and ERY groups were 14, 17, and 17 h, respectively). Subsequently, the pneumococcal culture supernatants were added at 50% to THP-1-derived macrophage cultures. The results in Fig. 1A show that the supernatants from AZM- and ERY-treated MRSP cells induced significantly lower tumor necrosis factor alpha (TNF-$\alpha$), interleukin-6 (IL-6), and IL-8 levels in THP-1-derived macrophages than did the supernatants from untreated MRSP cells. Macrolides

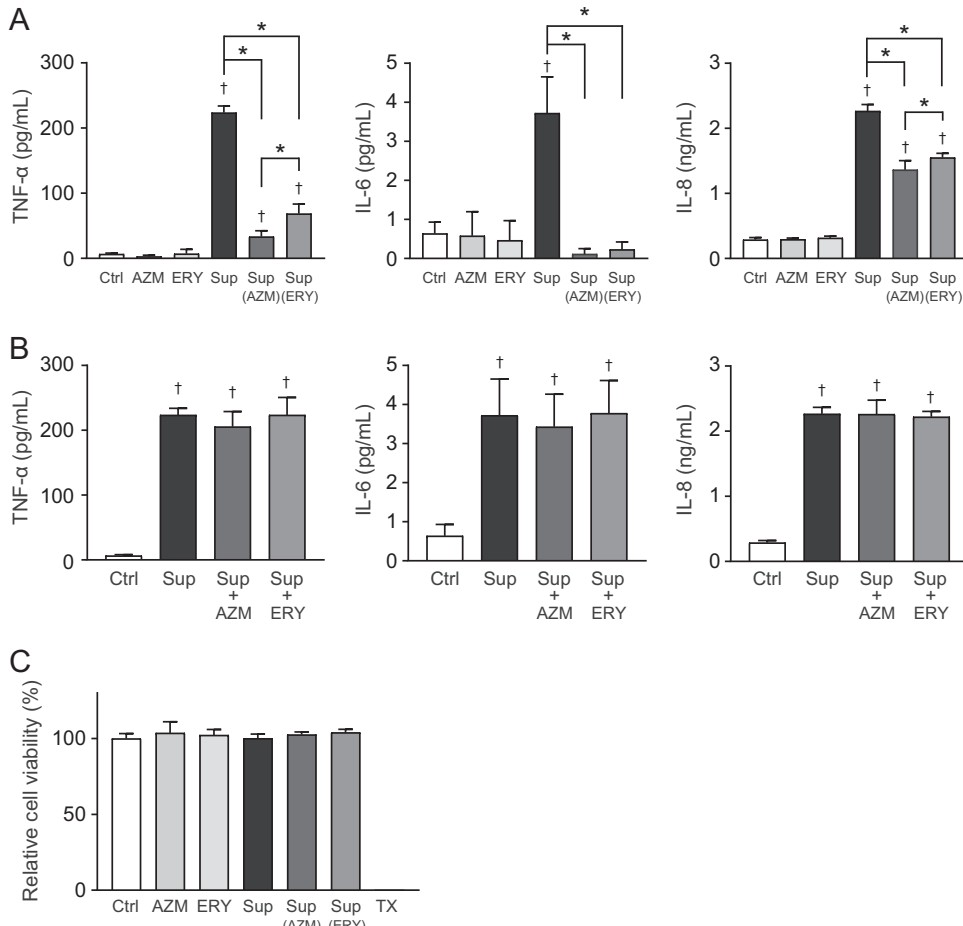

**FIG 1** Culture supernatants from macrolide-treated MRSP cells induce lesser inflammatory responses in THP-1-derived macrophages. MRSP strain NU4471 was inoculated into RPMI 1640 medium supplemented with 10% fetal bovine serum and cultured in the presence or absence of 5 $\mu$g/mL azithromycin (AZM) or erythromycin (ERY) until the cells reached the stationary phase. (A) THP-1-derived macrophages were exposed to 2-fold-diluted culture supernatant (Sup) from untreated or macrolide-treated MRSP cells or RPMI 1640 medium containing 2.5 $\mu$g/mL macrolides (AZM or ERY) for 9 h. TNF-$\alpha$, IL-6, and IL-8 levels in the macrophage culture supernatants were determined using ELISA. (B) THP-1-derived macrophages were exposed to supernatant from untreated MRSP cells in the presence or absence of AZM or ERY for 9 h. TNF-$\alpha$, IL-6, and IL-8 levels in the macrophage culture supernatants were determined using ELISA. (C) The MTT assay was performed to quantify the viability of THP-1-derived macrophages. Data represent the mean values $\pm$ standard deviations (SD) from quadruplicate experiments and were evaluated using one-way analysis of variance with Tukey's multiple-comparison test. †, significant difference compared with the control at $P < 0.05$; *, significant difference between the indicated groups at $P < 0.05$. Ctrl, control; MRSP, macrolide-resistant *S. pneumoniae*; MTT, thiazolyl blue tetrazolium bromide; TX, Triton X-100.

inhibit bacterial protein biosynthesis and show anti-inflammatory effects in host cells (17). Therefore, we mixed the supernatants from macrolide-untreated MRSP cells with AZM or ERY and added them to the THP-1-derived macrophage cultures. The results in Fig. 1B show that macrolides did not significantly alter the production of these cytokines in pneumococcal-supernatant-treated THP-1-derived macrophages. Additionally, the MTT [3-(4,5-dimethylthiazol-2-yl)-2,5-diphenyltetrazolium bromide] assay showed that neither macrolides nor pneumococcal supernatants decreased the viability of THP-1-derived macrophages (Fig. 1C). These findings suggest that AZM and ERY may decrease the release of pneumococcal proinflammatory components rather than exert immunosuppressive effects on macrophages under our experimental conditions.

**Supernatants from macrolide-treated MRSP cells induce significantly less NF-$\kappa$B activation in HEK293 cells expressing TLR2 and NOD2.** Previous studies have reported that different pattern recognition receptors (PRRs) recognize Gram-positive bacteria. Among these PRRs, *S. pneumoniae* activates Toll-like receptor 2 (TLR2), TLR4,

TLR9, and nucleotide-binding oligomerization domain 2 (NOD2) and induces NF-$\kappa$B-dependent expression of proinflammatory genes (18). To determine which pneumococcal components are affected by macrolide treatment, we used HEK-Blue cells expressing human TLR2 (hTLR2), hTLR4, hTLR9, or hNOD2 as NF-$\kappa$B reporter cells. HEK-Blue cells release secreted alkaline phosphatase (SEAP) into the cell culture medium in an NF-$\kappa$B-dependent manner. Therefore, the activation of PRRs by pneumococcal supernatants was assessed by measuring the SEAP activity. Notably, pneumococcal supernatants yielded robust SEAP activity in HEK-Blue cells expressing hTLR2 and hNOD2 (Fig. 2A and B), whereas neither hTLR4 nor hTLR9 was activated by the supernatants (Fig. 2C and D). HEK-Blue null2 cells, which were only transfected with the SEAP reporter gene but were without these PRRs, were unresponsive to pneumococcal supernatants (Fig. 2E). Additionally, supernatants from AZM- and ERY-treated MRSP cells induced significantly less SEAP activity in HEK-Blue cells expressing hTLR2 and hNOD2 than did supernatants from untreated MRSP cells (Fig. 2A and B). To assess whether macrolides exerted anti-inflammatory effects on HEK-Blue cells, we mixed the supernatants from macrolide-untreated MRSP cells with AZM or ERY and added them to the cultures. The results in Fig. 2F to J show that macrolides did not significantly alter SEAP activity in pneumococcal-supernatant-treated HEK-Blue cells. These data suggest that AZM and ERY decreased the release of TLR2 and NOD2 agonists from pneumococcal cells.

Although the pneumococcal supernatant did not activate NF-$\kappa$B via hTLR4 (Fig. 2C), the pore-forming toxin PLY, which is released extracellularly by autolysis (19), is known to induce NF-$\kappa$B activation via TLR4 (20). Additionally, AZM and ERY were previously demonstrated to decrease the release of PLY from MRSP (16). Therefore, to examine whether macrolide-induced inhibition of PLY release is involved in the decrease in NF-$\kappa$B activation, we quantified the PLY levels in the supernatant of the macrolide-treated MRSP cultures using a hemolytic activity assay, as PLY is responsible for pneumococcus-associated hemolysis (21). Contrary to previous studies, the RPMI medium-based pneumococcal supernatants used in this study did not show any hemolytic activity, whereas tryptic soy broth-based pneumococcal supernatants did induce hemolysis (Fig. S1A in the supplemental material). We then quantified the pneumococcal DNA levels in the bacterial supernatant using real-time PCR, as pneumococcal autolysis causes PLY and pneumococcal DNA leakage. Quantifying extracellular DNA enables the estimation of pneumococcal autolytic activity (22). The results in Fig. S1B show that RPMI medium-based supernatants contained markedly less extracellular DNA than tryptic soy broth-based supernatants. These findings suggest that the *S. pneumoniae* strain used in this study seldom induces autolysis in RPMI 1640 medium during its growth, leading to the absence of PLY in the supernatant. Therefore, the proinflammatory effects of PLY were negligible under these culture conditions.

**Macrolide-treated MRSP cells release significantly less peptidoglycan into the supernatant.** NOD2 senses pneumococcal peptidoglycan and induces NF-$\kappa$B activation (23), while TLR2 is considered to recognize pneumococcal cell wall components, such as peptidoglycan, lipoteichoic acid (LTA), and lipoproteins (24–26). Therefore, we hypothesized that macrolide treatment might inhibit the release of peptidoglycan into the supernatant of MRSP. We performed the silkworm larva plasma test, which detects peptidoglycan by activating the prophenoloxidase cascade (27). The results in Fig. 3 show that the concentrations of peptidoglycan in the supernatants from AZM- and ERY-treated MRSP cells were significantly lower than the concentration in supernatant from untreated MRSP cells.

**Macrolide-treated MRSP cells induce lesser inflammatory responses in THP-derived macrophages.** These findings prompted us to elucidate whether macrolide treatment alters the proinflammatory activity in pneumococcal cells. Macrolide-treated and untreated MRSP cells were killed by ethanol treatment and used in the proinflammatory activity assay. The results in Fig. 4A show that TNF-$\alpha$, IL-6, and IL-8 levels were slightly but significantly decreased in THP-1-derived macrophages stimulated with AZM- or ERY-treated MRSP cells compared to the levels in macrophages stimulated

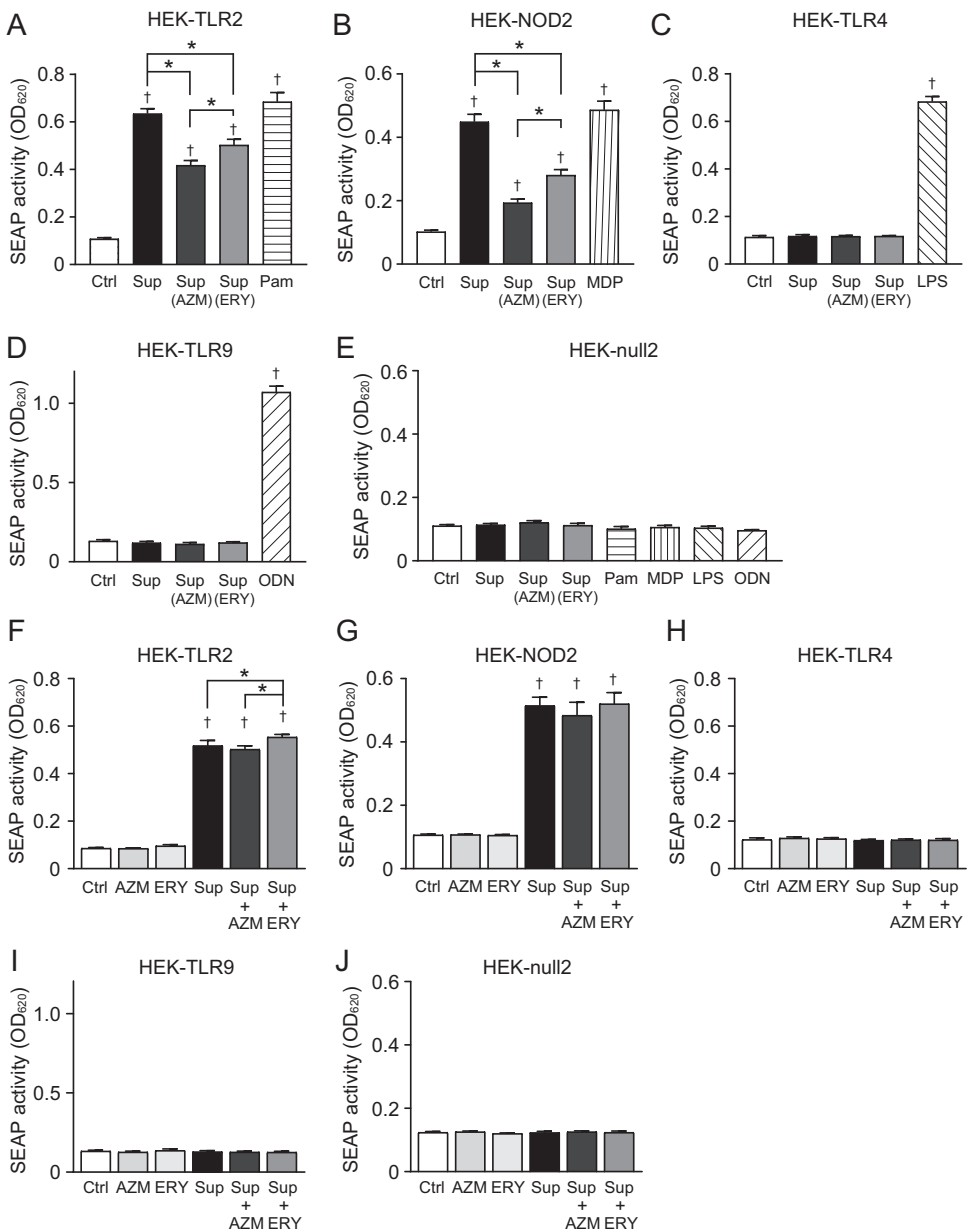

**FIG 2** Culture supernatants from macrolide-treated MRSP cells induce less TLR2 and NOD2 activation. MRSP strain NU4471 was grown in the presence or absence of 5 $\mu$g/mL azithromycin (AZM) or erythromycin (ERY) until it reached the stationary phase. (A to E) HEK-Blue hTLR-2 cells (A), HEK-Blue hNOD2 cells (B), HEK-Blue hTLR4 cells (C), HEK-Blue hTLR9 cells (D), and HEK-Blue null2 negative-control cells (E) were incubated with the culture supernatant (Sup) from untreated or macrolide-treated MRSP cells for 12 to 20 h (final concentrations of macrolides in macrolide-treated groups, 0.5 $\mu$g/mL). (F to J) HEK-Blue hTLR-2 cells (F), HEK-Blue hNOD2 cells (G), HEK-Blue hTLR4 cells (H), HEK-Blue hTLR9 cells (I), and HEK-Blue null2 negative-control cells (J) were exposed to supernatant from untreated MRSP cells in the presence or absence of 0.5 $\mu$g/mL AZM or ERY for 12 h. As controls, cells were also exposed to RPMI 1640 medium containing 0.5 $\mu$g/mL AZM or ERY. (A to J) Secreted alkaline phosphatase (SEAP) levels were quantified using spectrophotometry at 620 nm. Data represent the mean values $\pm$ SD from quintuplicate experiments and were evaluated using one-way analysis of variance with Tukey's multiple-comparison test. †, significant difference compared with the control at $P < 0.05$; *, significant difference between the indicated groups at $P < 0.05$. Ctrl, control; LPS, lipopolysaccharide; MDP, muramyl dipeptide; MRSP, macrolide-resistant *S. pneumoniae*; ODN, CpG ODN; Pam, Pam3CSK4.

with untreated MRSP cells. None of these stimuli altered the viability of macrophages (Fig. 4B). The SEAP activity assay using HEK-Blue cells revealed that ethanol-killed *S. pneumoniae* cells induced NF-$\kappa$B activation via TLR2 (Fig. 5A). Additionally, AZM- or ERY-treated MRSP cells induced significantly less SEAP activity in HEK-Blue cells

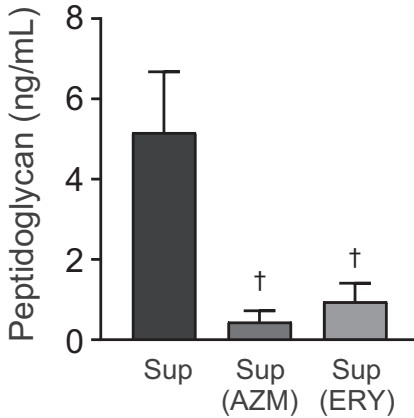

**FIG 3** Macrolide treatment significantly decreases the peptidoglycan concentration in MRSP culture supernatant. MRSP strain NU4471 was grown in the presence or absence of 5 $\mu$g/mL azithromycin (AZM) or erythromycin (ERY) until it reached the stationary phase. The peptidoglycan concentrations in the pneumococcal culture supernatants (Sup) were determined. Data represent the mean values $\pm$ SD from quadruplicate experiments and were evaluated using one-way analysis of variance with Tukey's multiple-comparison test. †, significant difference compared with the control at $P < 0.05$. MRSP, macrolide-resistant *S. pneumoniae*.

expressing hTLR2 than did untreated MRSP cells. However, HEK-Blue null2 and HEK-Blue cells expressing hNOD2, hTLR4, and hTLR9 were unresponsive to pneumococcal cells (Fig. 5B to E). These findings suggest that macrolide treatment decreases the expression of pneumococcal components that activate TLR2 in MRSP cells.

**Macrolide treatment disrupts the balance of the transcription of peptidoglycan synthesis-, lipoteichoic acid synthesis-, and lipoprotein synthesis-related genes in MRSP cells.** Peptidoglycan biosynthesis is catalyzed by Mur enzymes (28). Additionally, several proteins and enzymes, such as MraY and GatD, are involved in its biosynthesis.

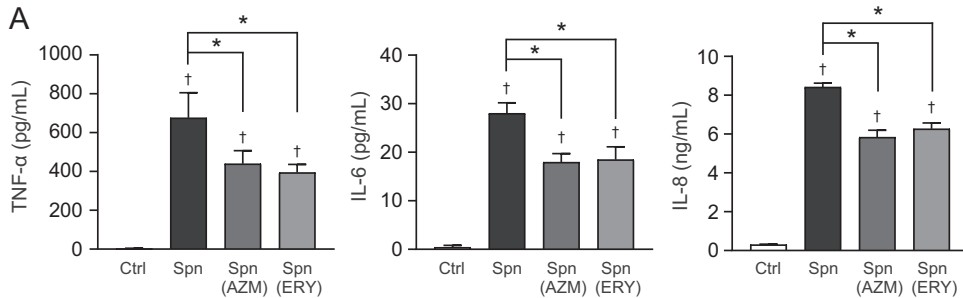

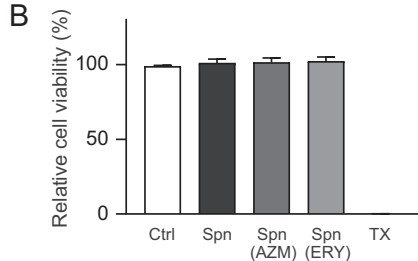

**FIG 4** Treatment of MRSP cells with macrolides induces lesser inflammatory responses in THP-derived macrophages. MRSP strain NU4471 was grown in the presence or absence of 5 $\mu$g/mL azithromycin (AZM) or erythromycin (ERY) until it reached the stationary phase. (A) THP-1-derived macrophages were stimulated with ethanol-killed *S. pneumoniae* cells (Spn) for 9 h. TNF-$\alpha$, IL-6, and IL-8 levels in macrophage culture supernatants were determined using ELISA. (B) The MTT assay was performed to quantify the viability of THP-1-derived macrophages. Data represent the mean values $\pm$ SD from quintuplicate experiments and were evaluated using one-way analysis of variance with Tukey's multiple-comparison test. †, significant difference compared with the control at $P < 0.05$; *, significant difference between the indicated groups at $P < 0.05$. Ctrl, control; MRSP, macrolide-resistant *S. pneumoniae*; MTT, thiazolyl blue tetrazolium bromide; TX, Triton X-100.

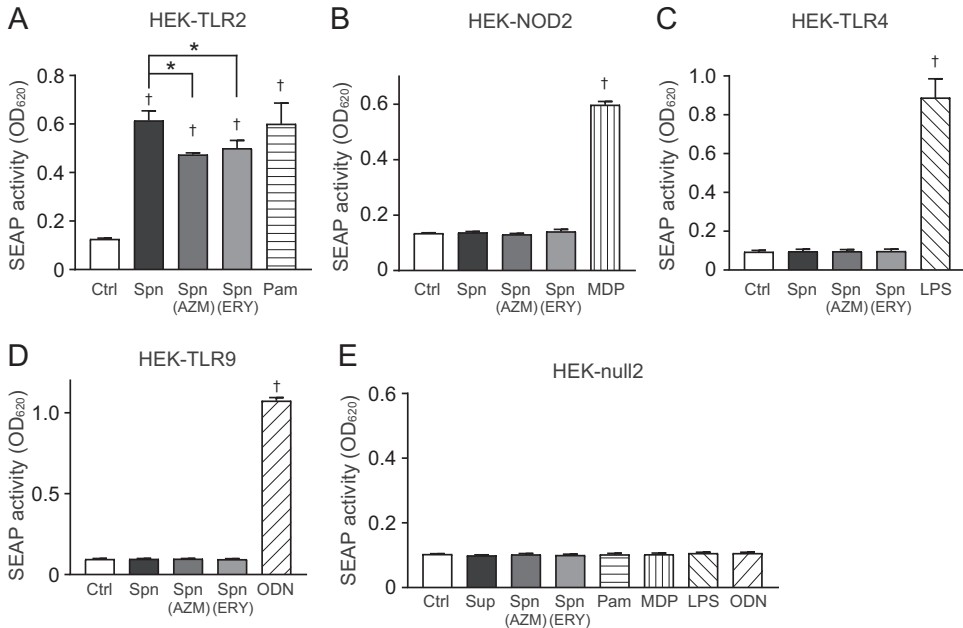

**FIG 5** Treatment of bacterial cells with macrolides induces less TLR2 activation. MRSP strain NU4471 was grown in the presence or absence of 5 $\mu$g/mL azithromycin (AZM) or erythromycin (ERY) until it reached the stationary phase. Pneumococcal cells were killed by treatment with 70% ethanol. (A) HEK-Blue hTLR-2 cells were stimulated with ethanol-killed *S. pneumoniae* (Spn) for 12 h. (B) HEK-Blue hNOD2 cells were stimulated with Spn for 18 h. (C) HEK-Blue hTLR4 cells were stimulated with Spn for 20 h. (D) HEK-Blue hTLR9 cells were stimulated with Spn for 20 h. (E) HEK-Blue null2 negative-control cells were stimulated with Spn for 17 h. (A to E) Secreted alkaline phosphatase (SEAP) levels were quantified using spectrophotometry at 620 nm. Data represent the mean values $\pm$ SD from quintuplicate experiments and were evaluated using one-way analysis of variance with Tukey's multiple-comparison test. †, significant difference compared with the control at $P < 0.05$; *, significant difference between the indicated groups at $P < 0.05$. Ctrl, control; LPS, lipopolysaccharide; MRSP, macrolide-resistant *S. pneumoniae*; MDP, muramyl dipeptide; ODN, CpG ODN; Pam, Pam3CSK4.

As our previous findings indicated that macrolide treatment affects the transcription of various pneumococcal genes, such as *ply* and *lytA* (15), we hypothesized that macrolide treatment also affects the transcription of peptidoglycan synthesis-related genes. Real-time PCR analysis showed that both AZM and ERY significantly decreased the transcription of *murA1*, *murB*, *murD*, *murE*, *murG*, *murT*, and *gatD*, whereas macrolides increased the transcription of *murC* and *murM* (Fig. 6). The transcription of *murF* and *mraY* was not significantly affected by macrolides.

Although several studies have demonstrated that bacterial peptidoglycan activates innate immune responses via TLR2 (24–26, 29, 30), Travassos et al. reported that peptidoglycan is not a TLR2 agonist but a NOD2 agonist (31). Instead, TLR2 stimulation by the cell wall components of Gram-positive bacteria is mediated by LTA. Therefore, although a macrolide-mediated reduction in the NOD2-stimulating activity of pneumococcal supernatants (Fig. 2B) may be caused by the reduction in peptidoglycan synthesis and release, we hypothesized that macrolides might also decrease the expression of LTA, resulting in a reduction in the TLR2-stimulating activity of pneumococcal cells and supernatants, as shown by the results in Fig. 2A and 5A. Therefore, we then attempted to determine the LTA levels in pneumococcal culture supernatants. However, the commercial LTA enzyme-linked immunosorbent assay (ELISA) kit could not detect pneumococcal LTA (Fig. S2).

Although the structure of LTA differs from that of wall teichoic acid (WTA) in most bacterial species, pneumococcal LTA and WTA are produced in the same biosynthetic pathway (32). At least 16 known and hypothetical genes contribute to the biosynthesis of the teichoic acid (TA) repeating units (32). We examined the transcriptional levels of *tarI*, *tarJ*, *licA*, *licB*, *licC*, and *licD1*, which are essential genes for TA biosynthesis, using real-time PCR. As shown by the results in Fig. 7, treatment with both AZM and ERY significantly decreased the transcription of *tarI*, *tarJ*, *licA*, and *licC*, whereas the transcription of

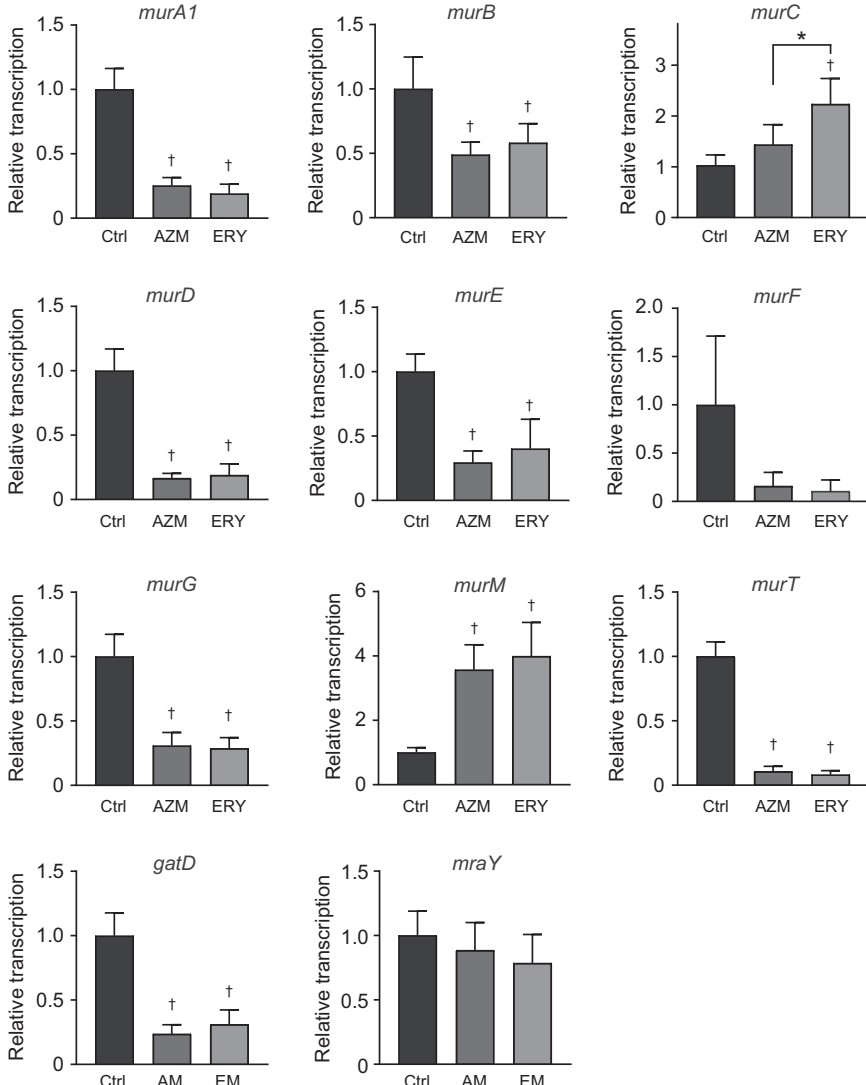

**FIG 6** Treatment of MRSP cells with macrolides affects the balance of peptidoglycan synthesis-related gene transcription. MRSP strain NU4471 was cultured in the presence or absence of 5 μg/mL azithromycin (AZM) or erythromycin (ERY) for 2 h. Real-time PCR was performed to quantify the transcription of genes encoding peptidoglycan synthesis-related molecules in MRSP cells. The relative quantities of target mRNAs were normalized to the relative quantity of 16S rRNA mRNA. Data represent the mean values ± SD from quadruplicate experiments and were evaluated using one-way analysis of variance with Tukey's multiple-comparison test. †, significant difference compared with the control at $P < 0.05$; *, significant difference between the indicated groups at $P < 0.05$. Ctrl, control; MRSP, macrolide-resistant *S. pneumoniae*.

*licD1* was decreased by AZM treatment alone. The transcription of *licB* was not significantly altered by the macrolides.

Pneumococcal lipoproteins are also major TLR2 ligands (26). Two lipoprotein biosynthetic enzymes, lipoprotein diacylglyceryl transferase (Lgt) and lipoprotein signal peptidase (Lsp), are essential in Gram-positive bacteria (33). Therefore, we assessed the effects of macrolides on the transcription of these enzymes. Although only a slight decrease in *lgt* gene transcription was observed in AZM-treated MRSP cells, transcription of the gene decreased by 55% in ERY-treated MRSP cells (Fig. 8A). In contrast, AZM decreased *lsp* transcription by 35%, whereas ERY did not. Previous studies reported that Triton X-114 phase partitioning extracts lipoproteins from bacterial cells (34, 35), including *S. pneumoniae* cells (36). We investigated the expression of lipoproteins in macrolide-treated MRSP cells using Coomassie blue staining of Triton X-114

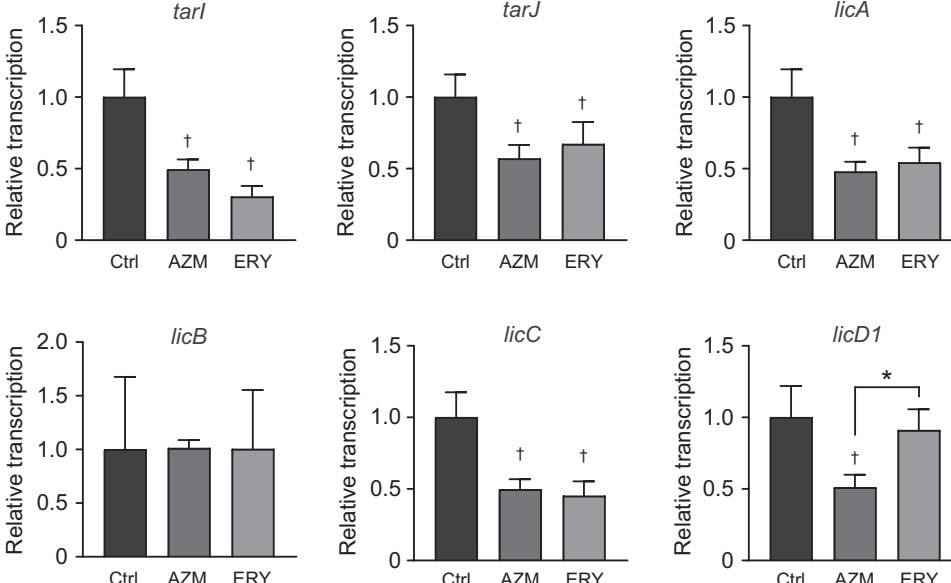

**FIG 7** Treatment of MRSP cells with macrolides decreased the transcription of the genes encoding lipoteichoic acid synthesis-related molecules. MRSP strain NU4471 was cultured in the presence or absence of 5 μg/mL azithromycin (AZM) or erythromycin (ERY) for 2 h. Real-time PCR was performed to quantify the transcription of the genes encoding lipoteichoic acid synthesis-related molecules in MRSP cells. The relative quantities of target mRNAs were normalized to the relative quantity of 16S rRNA mRNA. Data represent the mean values ± SD from quadruplicate experiments and were evaluated using one-way analysis of variance with Tukey's multiple-comparison test. †, significant difference compared with the control at $P < 0.05$; *, significant difference between the indicated groups at $P < 0.05$. Ctrl, control; MRSP, macrolide-resistant *S. pneumoniae*.

extracts. The results in Fig. 8B show that treatment with both AZM and ERY decreased the intensity of lipoproteins in MRSP cells. Additionally, a 37-kDa lipoprotein, one of the major bands detected in macrolide-untreated MRSP cells, was almost absent in the Triton X-114 extract from macrolide-treated MRSP cells.

As macrolides downregulate the transcription of multiple genes, we hypothesized that they induce morphological changes in pneumococcal cells. However, transmission

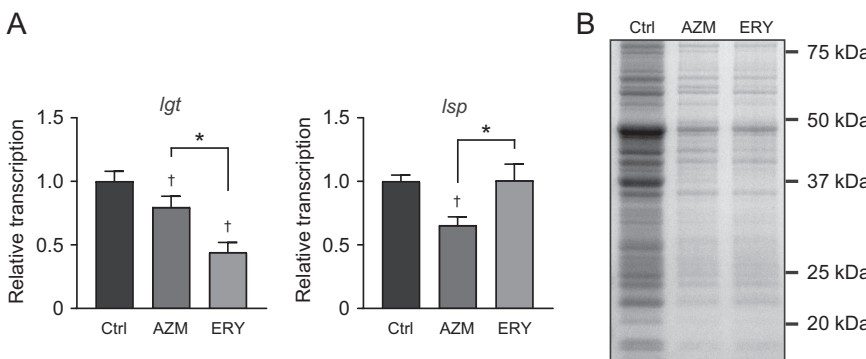

**FIG 8** Treatment of MRSP cells with macrolides downregulated the transcription of the genes encoding lipoprotein synthesis-related enzymes and decreased lipoprotein expressions in MRSP cells. (A) MRSP strain NU4471 was cultured in the presence or absence of 5 μg/mL azithromycin (AZM) or erythromycin (ERY) for 2 h. Real-time PCR was performed to quantify the transcription of the genes encoding lipoprotein synthesis-related enzymes in MRSP cells. The relative quantities of target mRNAs were normalized to the relative quantity of 16S rRNA mRNA. Data represent the mean values ± SD from quadruplicate experiments and were evaluated using one-way analysis of variance with Tukey's multiple-comparison test. †, significant difference compared with the control at $P < 0.05$; *, significant difference between the indicated groups at $P < 0.05$. (B) MRSP strain NU4471 was grown in the presence or absence of 5 μg/mL AZM or ERY until it reached the stationary phase. Coomassie blue-stained SDS-PAGE of Triton X-114 extracts prepared from these MRSP cells is shown. Ctrl, control; MRSP, macrolide-resistant *S. pneumoniae*.

Untreated                               AZM                               ERY

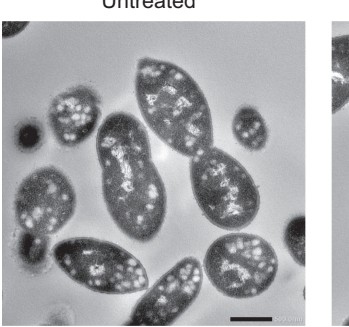 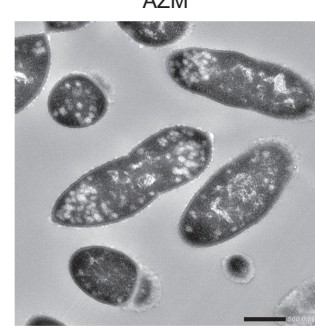 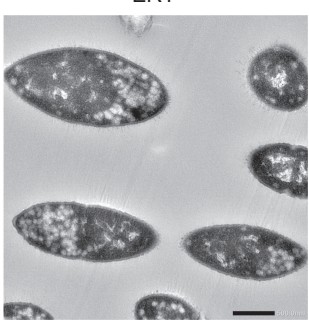

**FIG 9** Transition electron microscopy (TEM) analysis of macrolide-treated *S. pneumoniae* cells. MRSP strain NU4471 was grown in the presence or absence of 5 $\mu$g/mL azithromycin (AZM) or erythromycin (ERY) until it reached the stationary phase. Representative TEM images of untreated and macrolide-treated MRSP cells are shown. Scale bars = 500 nm.

electron microscopy (TEM) revealed that macrolides did not cause morphological changes (Fig. 9).

**Exposure of other clinical MRSP isolates to macrolides reduces the proinflammatory activity of the pneumococcal supernatants.** To assess whether these effects of macrolides are generalized to other MRSP isolates, we used the clinical MRSP strain KM2412, collected from the nasopharynx of patients with acute otitis media (3). Consistent with the findings in MRSP strain NU4471, the supernatants from AZM- and ERY-treated MRSP strain KM2412 cells induced significantly lower levels of TNF-$\alpha$, IL-6, and IL-8 in THP-1-derived macrophages than did the supernatant from untreated MRSP cells (Fig. 10A). However, the addition of macrolides to the supernatant from macrolide-untreated MRSP strain KM2412 did not alter the production of these cytokines in THP-1-derived macrophages. The results in Fig. 10B show that the supernatant from strain KM2412 did not affect the viability of the macrophages. Additionally, macrolide treatment significantly decreased the release of peptidoglycan from the MRSP strain KM2412 into the supernatant (Fig. 10C). The results in Fig. 10D show that treatment with both AZM and ERY decreased the intensity of lipoproteins in MRSP strain KM2412 cells. These data show that the macrolides also induced a decrease in the proinflammatory activity of *S. pneumoniae* strain KM2412.

## DISCUSSION

In this study, we found that the supernatants from macrolide-treated MRSP cells induced significantly less TLR2 and NOD2 activation in HEK-Blue cells than did the supernatant from untreated MRSP cells. A possible mechanism is the macrolide-induced disruption of the balance of peptidoglycan synthesis-related gene transcription. Additionally, macrolide treatment significantly decreased the transcription of several LTA and lipoprotein synthesis-related genes in MRSP cells. Our previous study also demonstrated that macrolide treatment downregulated the transcription of the *gapdh*, *dnaK*, and *ply* genes in MRSP (15). Our findings indicate that macrolides alter the transcription of multiple pneumococcal genes, including those essential for bacterial cell homeostasis, without affecting bacterial survival. Consequently, macrolides may lead to a delayed or decreased release of bacterial ligands for innate immune receptors, such as peptidoglycan, LTA, and lipoprotein, resulting in the decreased inflammatory activity of pneumococcal supernatants.

Peptidoglycan is a specific component of the bacterial cell wall outside the cytoplasmic membrane. The biosynthesis of pneumococcal peptidoglycan is reportedly initiated by the synthesis of UDP-Mur*N*Ac by the MurA and MurB enzymes. Thereafter, MurC, MurD, MurE, and MurF (ATP-dependent ligases) catalyze the sequential addition of amino acids to UDP-Mur*N*Ac (28). The MurG and MraY enzymes catalyze the reactions in the final stages of cytoplasmic peptidoglycan synthesis, generating the main peptidoglycan building block, lipid II (37). The MurT/GatD enzyme complex amidates lipid II (38). MurM catalyzes the tRNA-dependent addition of a dipeptide to the lysine residue of a

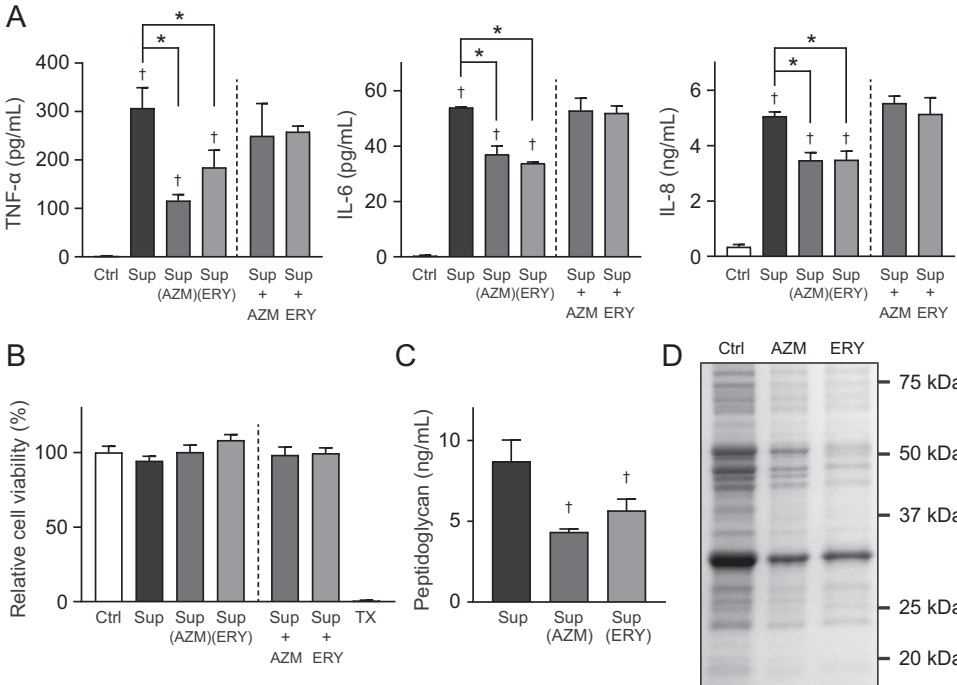

**FIG 10** Exposure of other clinical MRSP isolates to macrolides results in decreased proinflammatory activity of the culture supernatants. MRSP strain KM2412 was inoculated into RPMI 1640 medium supplemented with 10% fetal bovine serum and cultured in the presence or absence of 4 μg/mL azithromycin (AZM) or erythromycin (ERY) until it reached the stationary phase. (A) THP-1-derived macrophages were exposed to the culture supernatant (Sup) from untreated or macrolide-treated MRSP cells for 9 h. TNF-α, IL-6, and IL-8 levels in the macrophage culture supernatants were determined using ELISA. (B) The MTT assay was performed to quantify the viability of THP-1-derived macrophages. (C) Peptidoglycan concentrations in pneumococcal supernatants were determined. (A to C) Data represent the mean values ± SD from quadruplicate experiments and were evaluated using one-way analysis of variance with Tukey's multiple-comparison test. †, significant difference compared with the control at $P < 0.05$; *, significant difference between the indicated groups at $P < 0.05$. (D) Coomassie blue-stained SDS-PAGE gel of the Triton X-114 extract prepared from MRSP cells. Ctrl, control; MRSP, macrolide-resistant *S. pneumoniae*; MTT, thiazolyl blue tetrazolium bromide; TX, Triton X-100.

peptidoglycan precursor and plays a key role in peptidoglycan cross-linking (39). The synthesized peptidoglycan is remodeled during bacterial growth, leading to the release of peptidoglycan fragments (40). In this study, 7 of 11 genes encoding peptidoglycan-synthetic enzymes were significantly downregulated by AZM and ERY (Fig. 11A). These findings suggest that macrolides disturb the homeostatic balance of the peptidoglycan biosynthesis pathway, which may lead to reduced peptidoglycan release. Since peptidoglycan-cleaving enzymes, such as peptidoglycan hydrolases, also play a role in the release of peptidoglycan fragments during cell growth (41), further studies are needed to evaluate the transcriptional regulation of these enzymes by macrolides.

NOD1 and NOD2 are intracellular pattern recognition receptors that respond to a variety of signaling molecules, including peptidoglycan fragments. NOD1 detects the peptidoglycan motif γ-D-glutamyl-*meso*-diaminopimelic acid, which is found primarily in Gram-negative bacteria (42), whereas NOD2 detects muramyl dipeptide, which is ubiquitously present in both Gram-positive and Gram-negative bacteria (43). In addition, pneumococcal peptidoglycan fragments are reportedly recognized by NOD2, leading to NF-κB activation and proinflammatory cytokine production (23). Together, the present findings indicate that macrolide-induced inhibition of peptidoglycan release results in impaired NF-κB activation and decreased production of proinflammatory cytokines via NOD2 in HEK-Blue cells and THP-derived macrophages, respectively.

TLR2 was first shown to recognize peptidoglycan in 1999 (24, 44). Although many studies have corroborated this observation, it has been since reported that TLR2 is not stimulated by the synthetic peptidoglycan structure (45). Additionally, highly purified

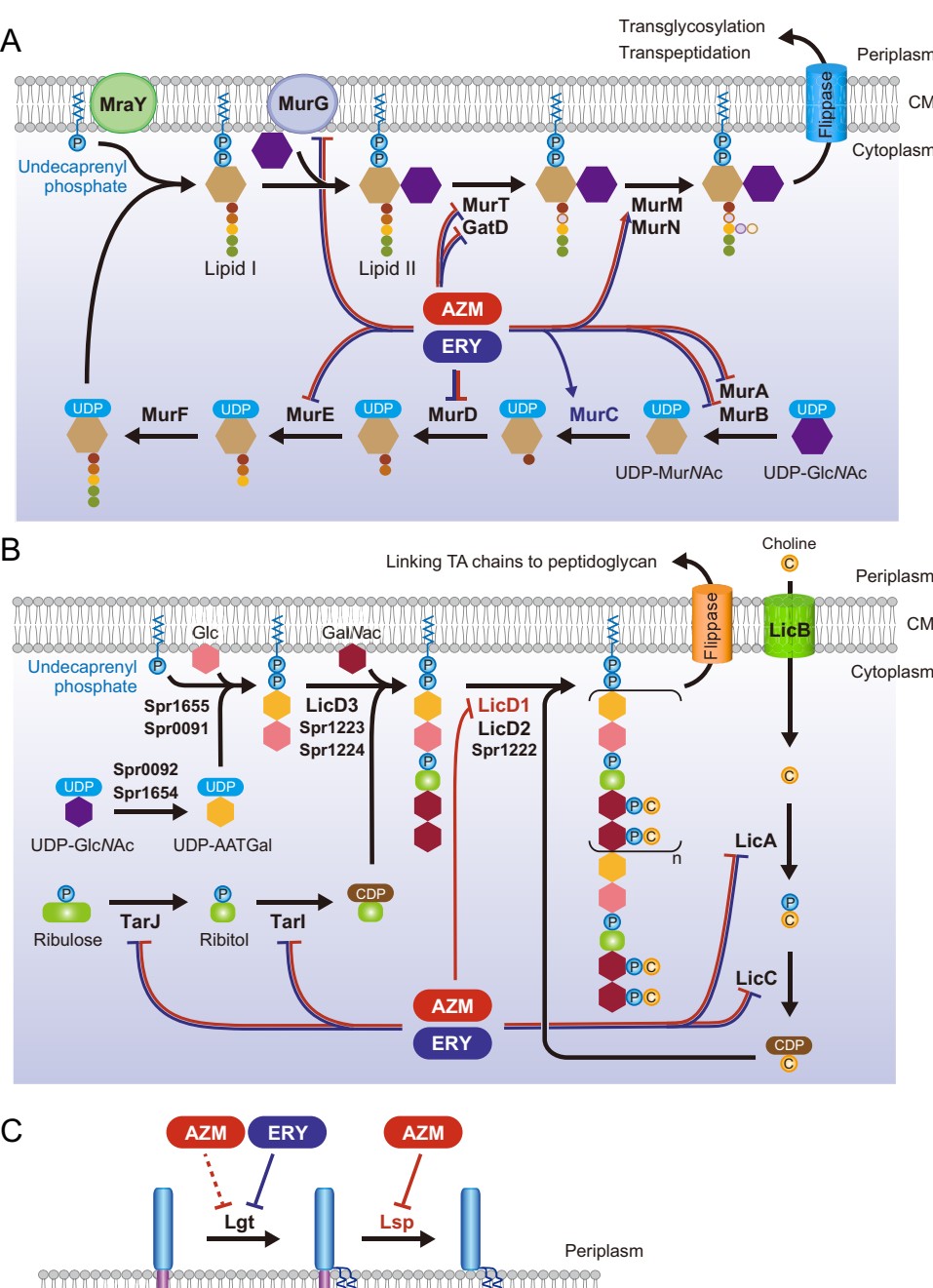

**FIG 11** Scheme of macrolide-induced alterations in the transcription of pneumococcal genes encoding cell wall synthesis-related molecules. (A) Transcriptional alterations in peptidoglycan synthesis-related gene transcription in macrolide-treated MRSP cells. (B) Transcriptional alterations in the proposed lipoteichoic acid synthesis-related gene transcription in macrolide-treated MRSP cells. Although the pathway applies to *S. pneumoniae* generally, the gene/protein names are from *S. pneumoniae* strain R6. (C) Transcriptional alterations in lipoprotein synthesis-related gene transcription in MRSP cells treated with macrolides. AATGal, 2-acetamido-4-amino-2,4,6-trideoxygalactose; AZM, azithromycin; CM, cytoplasmic membrane; ERY, erythromycin; Gal*N*Ac, *N*-acetylgalactosamine; Glc, glucose; Glc*N*Ac, *N*-acetylglucosamine; MRSP, macrolide-resistant *S. pneumoniae*; Mur*N*Ac, *N*-acetylmuramic acid; P, phosphate residue; TA, teichoic acid.

peptidoglycan from *S. pneumoniae* is not sensed by TLR2, TLR2/1, or TLR2/6 (31). We found that macrolide treatment decreased the release of peptidoglycan fragments into the pneumococcal supernatant, which may not account for the decrease in TLR2 activation in the supernatant-treated HEK-Blue cells. Several studies have reported that *S. pneumoniae* LTA activates NF-$\kappa$B via TLR2 (31, 46). Therefore, we examined the effect of macrolides on the transcription of six essential genes encoding LTA-synthetic enzymes and found that five genes were significantly downregulated by AZM or ERY or both. The schematic in Fig. 11B shows the proposed TA biosynthesis pathway and the macrolide-induced transcriptional alterations of the genes in *S. pneumoniae*.

Tomlinson et al. reported that TLR2-mediated inflammatory responses in *S. pneumoniae* are highly dependent on the surface expression of lipoproteins (26). Pneumococcal lipoproteins, as well as those in other bacteria, are initially translated into the cytoplasm as preprolipoproteins containing an N-terminal signal peptide. After export across the cell membrane, preprolipoproteins are anchored to the cell membrane by adding diacylglyceryl moieties via the lipoprotein diacylglyceryl transferase Lgt. The lipoprotein signal peptidase Lsp cleaves the signal peptide of prolipoproteins, which results in the maturation of lipoproteins (47). A surface lipoprotein-deficient $\Delta lgt$ pneumococcal mutant strain exhibited markedly less NF-$\kappa$B activation in HEK293 cells expressing TLR2 (26). Although the role of Lsp in *S. pneumoniae*-induced TLR2 activation is not fully understood, a $\Delta lsp$ mutant strain of *Streptococcus agalactiae* exhibited largely impaired activation of NF-$\kappa$B in HEK293 cells expressing TLR2 (48). These findings indicate that proper processing of lipoproteins, which is catalyzed by both Lgt and Lsp, is essential for TLR2-stimulatory activity. Additionally, growing bacteria release lipoproteins and the secreted forms of lipoproteins can induce proinflammatory cytokines (49). In this study, AZM and ERY mainly downregulated the transcription of *lsp* and *lgt*, respectively (Fig. 11C). Additionally, these macrolides induced marked reductions in lipoprotein levels in MRSP strains. Therefore, transcriptional downregulation of *lsp* and *lgt* may be one of the mechanisms involved in the macrolide-induced decrease in the proinflammatory activity of pneumococcal cells and supernatants via TLR2. Although we focused on the transcriptional regulation of two lipoprotein synthesis enzymes, *S. pneumoniae* was predicted to possess up to 37 lipoproteins (47). Further studies are required to assess the effects of macrolides on the transcription of individual lipoprotein genes.

Although pneumococcal supernatants induced NF-$\kappa$B activation in HEK-Blue cells via both TLR2 and NOD2, pneumococcal cells mainly activated TLR2 but not NOD2. The mechanisms by which bacterial peptidoglycan enters host cells and activates NOD2 receptors include (i) phagocytosis of bacteria and subsequent bacterial degradation, (ii) uptake of peptidoglycan fragments from bacterial outer membrane vesicles, and (iii) endocytosis, as well as some other routes (50). Since HEK-Blue cells are nonphagocytic, they do not take up peptidoglycan via the phagocytic route. Additionally, in this study, to avoid bacterial growth in the HEK-Blue culture, pneumococcal cells were treated with ethanol, followed by a wash in phosphate-buffered saline (PBS). Thus, the samples do not likely contain outer membrane vesicles or secreted forms of peptidoglycan fragments. Therefore, pneumococcal cells possibly activate cell surface TLR2 but not the intracellular innate immune receptor NOD2.

In conclusion, our findings suggest that macrolides downregulate the transcription of various proinflammatory-component-related genes in MRSP. Among these genes, transcriptional downregulation of peptidoglycan synthesis-related genes by macrolides results in a decreased release of peptidoglycan, leading to reduced activation of NOD2 in THP-1-derived macrophages and HEK-Blue cells. Additionally, transcriptional downregulation of LTA synthesis- and lipoprotein synthesis-related genes by macrolides is likely to contribute to the decreased activation of TLR2 in these cells. Our previous study demonstrated that oral administration of macrolides to mice intratracheally infected with MRSP resulted in decreased IL-6 protein levels in bronchoalveolar lavage fluid samples compared to the levels in samples from untreated infected control mice, without affecting the bacterial loads in the samples (15). Additionally, Yasuda et al.

reported that the administration of roxithromycin, a 14-membered macrolide that does not inhibit PLY production, increased the median survival time and retarded bacteremia in a murine pneumonia model with MRSP (51). Although the immunomodulatory effects of macrolides may partly contribute to their *in vivo* efficacy, these findings suggest that macrolides may also affect the transcription of proinflammatory-component-related genes in MRSP *in vivo* (15). Collectively, our previous and present findings provide additional explanations for the clinical benefits of macrolides.

## MATERIALS AND METHODS

**Reagents.** AZM and ERY were obtained from Tokyo Chemical Industry (Tokyo, Japan) and Fujifilm Wako Pure Chemical Corporation (Osaka, Japan), respectively. The macrolides were dissolved in ethanol and diluted with ultrapure distilled water (Thermo Fisher Scientific, Waltham, MA, USA). The MTT reagent for the cell viability assay was purchased from Merck KGaA (Darmstadt, Germany).

**Bacterial culture.** Two clinical macrolide-resistant pneumococcal strains, NU4471 (ERY MIC of >1,000 $\mu$g/mL) and KM2412 (ERY MIC of 12 $\mu$g/mL), which both harbor macrolide resistance genes *ermB* and *mefA*, were used in this study (15, 52). They were grown in tryptic soy broth (Becton, Dickinson, Franklin Lakes, NJ, USA) for 10 to 12 h under aerobic conditions at 37°C. The overnight cultures were inoculated into RPMI 1640 medium supplemented with 10% fetal bovine serum (FBS) and cultured in the presence or absence of 4 to 5 $\mu$g/mL AZM or ERY until bacterial growth reached the stationary phase (for strain NU4471, an $OD_{600}$ of 0.55, with incubation periods of the untreated control, AZM, and ERY treatment groups being 14, 17, and 17 h, respectively; for strain KM2412, an $OD_{600}$ of 0.1, with incubation periods of the untreated control, AZM, and ERY groups being 7, 14, and 15 h, respectively). The pneumococcal culture supernatants were obtained after centrifugation at 5,000 × *g* for 10 min, followed by filtration through a 0.22-$\mu$m-pore-size filter (GVS Filter Technology, Indianapolis, IN, USA). The supernatants were diluted 2-fold with fresh RPMI 1640 medium containing 10% FBS and used for subsequent THP-1 stimulation. The undiluted supernatants were used for other assays. The bacterial pellets obtained after centrifugation were treated with 70% ethanol for 10 min, which completely killed the organisms. Then, the pellets were washed with PBS and resuspended in PBS. The turbidity of the bacterial suspensions was measured spectrophotometrically at 600 nm (Bio-Rad, Hercules, CA, USA), and the suspensions were used for HEK-Blue cell stimulation.

**Cell preparation, culture, and stimulation.** The monocytic cell line THP-1 was maintained in RPMI 1640 medium supplemented with 10% FBS, 100 U/mL penicillin, and 100 $\mu$g/mL streptomycin at 37°C with 5% $CO_2$. For cytokine assays, the cells were incubated in a 24-well culture plate at a density of $3 \times 10^5$ cells/mL in a medium supplemented with 200 nM phorbol 12-myristate 13-acetate for 48 h to induce differentiation toward macrophage-like cells. The cells were washed extensively with RPMI 1640 medium and cultured further in medium without FBS for 12 h. Then, the cells were exposed to ethanol-killed pneumococci at a density of $6 \times 10^6$ cells/mL (multiplicity of infection of 20) or 2-fold-diluted pneumococcal culture supernatants from untreated or macrolide-treated MRSP cells for 9 h. Additionally, to assess the direct effect of macrolides on cytokine release, THP-1 cells were exposed to 2-fold-diluted culture supernatants from untreated MRSP cells in the presence or absence of 2.5 $\mu$g/mL AZM or ERY. Cell viability assays using the MTT reagent were performed as previously described (53).

**Cytokine assay.** The TNF-$\alpha$, IL-6, and IL-8 levels in the supernatants of THP-1 cultures were determined using commercially available ELISA kits (BioLegend, San Diego, CA, USA) according to the manufacturer's instructions.

**SEAP activity assay.** HEK-Blue null2, hTLR2, hTLR4, hTLR9, and hNOD2 cells were purchased from InvivoGen (San Diego, CA, USA). These cells were cultured at 37°C with 5% $CO_2$ in Dulbecco's modified Eagle's medium (DMEM; Fujifilm Wako Pure Chemical Corporation) supplemented with 10% FBS, 100 U/mL penicillin, 100 $\mu$g/mL streptomycin, and 100 $\mu$g/mL Normocin (InvivoGen) in the presence of selective antibiotics. The selection antibiotics used were 100 $\mu$g/mL Zeocin (InvivoGen) for HEK-Blue null2 cells, HEK-Blue selection (InvivoGen) for hTLR2 and hTLR4 cells, 10 $\mu$g/mL blasticidin (InvivoGen) and 100 $\mu$g/mL Zeocin for hTLR9 cells, and 30 $\mu$g/mL blasticidin and 100 $\mu$g/mL Zeocin for hNOD2 cells. These cell lines were suspended in HEK-Blue detection medium (InvivoGen), which contains a specific SEAP substrate that allows the colorimetric detection of SEAP activity, seeded at a density of $5 \times 10^4$ cells per 180 $\mu$L in 96-well plates, and then treated with 20 $\mu$L pneumococcal supernatant, ethanol-killed pneumococci ($5 \times 10^5$ cells per 20 $\mu$L; a multiplicity of infection of 10), 2 ng/mL Pam3CSK4 (TLR1/2 ligand; InvivoGen), 25 ng/mL muramyl dipeptide (NOD2 ligand; InvivoGen), 10 ng/mL lipopolysaccharide from *Escherichia coli* strain 055:B5 (TLR4 ligand; Merck KGaA), or 2.5 $\mu$g/mL CpG oligodeoxynucleotide 2006 (TLR9 ligand; InvivoGen) for 12 to 20 h. Additionally, to assess the direct effect of macrolides on SEAP activity, HEK-Blue cells were exposed to culture supernatants from untreated MRSP cells in the presence or absence of AZM or ERY. SEAP activity was measured using a Multiskan FC microplate photometer (Thermo Fisher Scientific) at 620 nm.

**Silkworm larva plasma test.** The peptidoglycan concentrations in the pneumococcal culture supernatant were determined using the SLP-HS single reagent set (Fujifilm Wako Pure Chemical Corporation). Briefly, the staphylococcal peptidoglycan standard and pneumococcal culture supernatant samples were diluted with endotoxin-free distilled water, mixed with SLP-HS reagent II solution, and incubated at 30°C. Melanin formation was evaluated every 5 min by measuring the absorbance at 450 nm for >60 min using a spectrophotometer. Then, the values were converted to report the peptidoglycan concentrations in the samples by interpolating from the standard curve data according to the manufacturer's instructions.

**TABLE 1** Primer sequences used during real-time PCR analyses

| Target | | Sequence (5'–3') |
| --- | --- | --- |
| murA1 | Forward | TATCCGTGGGGCAGGAACTA |
| murA1 | Reverse | AGCTTCAATGCGGTCTGGAA |
| murB | Forward | CTTGGCTTTTGGTTACCGCC |
| murB | Reverse | TGAGTTCACGTAGGTGCGTC |
| murC | Forward | GTGCCTTGGCCTTGATGTTG |
| murC | Reverse | AAGACCGCGTTGGGTAAAGT |
| murD | Forward | TGGTGAATCTGCAGCTCGTT |
| murD | Reverse | ACTTTGGGCAGCTGGATTGT |
| murE | Forward | TGGACTTGCTTGTCTCCGTC |
| murE | Reverse | TTCCATACGACCAGGAACGC |
| murF | Forward | CGTCCAAAAACAGCCATCGT |
| murF | Reverse | GGCGCTAAAAGCAAGGAACC |
| murG | Forward | TCATGTTGCGACTTCGTCCA |
| murG | Reverse | TTGGCCAAGCCCATAGACAG |
| murM | Forward | AGCGAGAGTTCCTTTAGCGG |
| murM | Reverse | AGCGAGCCGTTTCATACCAT |
| murT | Forward | CTTGGACGTGGAAGTACGCT |
| murT | Reverse | GACAGTGAGGGCAGTTGTCA |
| gatD | Forward | AAAGGTCGGTGAAGGGGTTC |
| gatD | Reverse | TACGAGAGAGGATAGGCCCG |
| mraY | Forward | CGCAAATTACAGGCCAGCAG |
| mraY | Reverse | CCCATTGTAGGAGTCCCAGC |
| tarI | Forward | TGGTGCTGACCGCAATACAA |
| tarI | Reverse | TGGACGAACAGAATCGTGGG |
| tarJ | Forward | CGTCCCAACTACATGGCTGT |
| tarJ | Reverse | GTTCCGGTCGGGTCAGAAAT |
| licA | Forward | GGGATTTGGCTGCCCTCTTT |
| licA | Reverse | ACCGGTGTTTGGTCACTCTC |
| licB | Forward | ATCAATCGCTTGCAACCAGC |
| licB | Reverse | ACTGCAAACAAGACCGTCCA |
| licC | Forward | AGCCATCATCTTAGCAGCGG |
| licC | Reverse | CTGAACCAAGGCTTTAGGGGT |
| licD1 | Forward | ACTGGAAAGCTTCAAACTGCTG |
| licD1 | Reverse | GAGGAGAAACCGGTCGAAGG |
| lgt | Forward | CTCAAAGTTTGGGGCGTTGG |
| lgt | Reverse | AAGTCGGTTGACGGTAGCTC |
| lsp | Forward | GGTTGTCGTGATAGGTGCCA |
| lsp | Reverse | AACAAAGCCCTGACTGACCC |

**Quantitative real-time PCR.** The MRSP strain NU4471 was inoculated into RPMI 1640 medium supplemented with 10% FBS and cultured until it reached the early exponential growth phase ($OD_{600}$ of 0.1). Thereafter, the bacterial cultures were incubated in the presence or absence of 5 $\mu$g/mL AZM or ERY for 2 h at 37°C, followed by centrifugation at 5,000 × $g$ for 10 min. The bacterial pellet was resuspended in 1 mL TRI reagent (Molecular Research Center, Inc., Cincinnati, OH, USA) and homogenized using a MagNA Lyser instrument (Roche Diagnostics, Basel, Switzerland) and lysing matrix B (0.1-mm silica spheres in 2-mL tube; MP Biomedicals, Solon, OH, USA), followed by RNA isolation using a Direct-zol RNA miniprep kit (Zymo Research, Irvine, CA, USA). RNA was reverse transcribed to cDNA using SuperScript VILO master mix (Thermo Fisher Scientific). Quantitative real-time PCR was carried out in a StepOne plus real-time PCR system (Thermo Fisher Scientific) using Thunderbird Next SYBR qPCR mix (Toyobo, Osaka, Japan) according to the manufacturer's instructions. The primers used for real-time PCR are listed in Table 1. The relative quantities of the target mRNA were normalized to the relative quantity of 16S rRNA mRNA.

**Triton X-114 lipoprotein extraction.** Triton X-114 extraction was performed following the Bordier method (34), with minor modifications (35). Briefly, MRSP strains NU4471 and KM2412 were inoculated into RPMI 1640 medium supplemented with 10% FBS and cultured in the presence or absence of 4 to 5 $\mu$g/mL AZM or ERY until bacterial growth reached the stationary phase. Five-milliliter amounts of the pneumococcal cultures were centrifuged at 5,000 × $g$ for 10 min, and the cell pellets were digested with 100 $\mu$L of 0.1% sodium deoxycholate (Dojindo Laboratories, Kumamoto, Japan) in PBS, followed by incubation at 37°C for 30 min. The samples were sonicated with three pulses of 10 s each, using a Vibra-Cell VC602 ultrasonicator (Sonics & Materials, Inc., Newtown, CT, USA). Thereafter, 800 $\mu$L of PBS and 100 $\mu$L of 10% (vol/vol) Triton X-114 (Merck KGaA) were added to the samples, which were then incubated at 4°C for 2 h, followed by centrifugation at 13,000 × $g$ for 10 min at 4°C to remove insoluble debris. The supernatants were transferred to fresh tubes and incubated at 37°C for 30 min to allow phase separation. Thereafter, the detergent-phase proteins were pelleted by centrifugation at room temperature. The detergent pellets were then washed with 1 mL of PBS at 4°C for 1 h, followed by incubation at 37°C for

30 min and centrifugation at 13,000 × $g$ for 10 min. The samples were resuspended in 100 $\mu$L SDS sample buffer and separated by SDS-PAGE, followed by Coomassie brilliant blue staining.

**TEM analysis.** MRSP strain NU4471 was inoculated into RPMI 1640 medium supplemented with 10% FBS and cultured in the presence or absence of 5 $\mu$g/mL AZM or ERY until bacterial growth reached the late exponential phase (OD$_{600}$ of 0.4; the incubation periods of the untreated control, AZM, and ERY groups were 12, 15, and 16 h, respectively), followed by centrifugation at 5,000 × $g$ for 10 min. Bacterial pellets were resuspended and fixed in 2% glutaraldehyde (Electron Microscopy Sciences, Hatfield, PA, USA) in 0.1 M phosphate buffer (pH 7.4) at 4°C. Then, the samples were treated with 0.5% potassium permanganate and post-fixed in 2% osmium tetraoxide (Heraeus Chemicals South Africa, Port Elizabeth, South Africa) for 2 h at 4°C. The specimens were dehydrated in graded ethanol and embedded in an epoxy resin (TAAB Laboratories Equipment Ltd., Aldermaston, Berkshire, UK). Ultrathin sections were obtained using an ultramicrotome, stained with 2% uranyl acetate for 15 min, and subjected to TEM (JEM-1400Flash; JEOL Ltd., Tokyo, Japan) observation at 100 kV. These analyses were performed by Filgen (Aichi, Japan).

**Statistical analysis.** Data were statistically analyzed by analysis of variance with Tukey's multiple-comparison test using GraphPad Prism software version 9.4.0 (GraphPad Software, Inc., La Jolla, CA, USA).

**Data availability.** All data described are contained within the manuscript.

## SUPPLEMENTAL MATERIAL

Supplemental material is available online only.
**SUPPLEMENTAL FILE 1**, PDF file, 0.2 MB.

## ACKNOWLEDGMENTS

We thank Osamu Kimura (Shiokaze Clinic, Niigata, Japan) and Kotobiken Medical Laboratories (Niigata, Japan) for providing MRSP strain KM2412. This work was supported by the Japan Society for the Promotion of Science (JSPS) KAKENHI (grants number JP20K09903, JP20H03858, JP22H03267, JP22K09923, and JP22K19614). The funders had no role in study design, data collection and interpretation, or the decision to submit the work for publication.

We declare no competing financial interests.

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
