## [Reviewer comments · Microbiology Spectrum]

Microbiology Spectrum

Macrolides decrease the pro-inflammatory activity of macrolide-resistant *Streptococcus pneumoniae*

Hisanori Domon, Satoru Hirayama, Toshihito Isono, Karin Sasagawa, Fumio Takizawa, Tomoki Maekawa, Katsunori Yanagihara, and Yutaka Terao

Corresponding Author(s): Yutaka Terao, Niigata University Graduate School of Medical and Dental Sciences

Review Timeline:

Submission Date:	January 16, 2023
Editorial Decision:	February 27, 2023
Revision Received:	April 21, 2023
Accepted:	April 25, 2023

Editor: Christopher LaRock

Reviewer(s): The reviewers have opted to remain anonymous.

Transaction Report:

DOI: <https://doi.org/10.1128/spectrum.00148-23>

February 27, 2023

Prof. Yutaka Terao
Niigata University Graduate School of Medical and Dental Sciences
Division of Microbiology and Infectious Diseases
5274 Gakkocho 2-ban-cho, Chuo-ku
Niigata 951-8514
Japan

Re: Spectrum00148-23 (Macrolides decrease the pro-inflammatory activity of macrolide-resistant *Streptococcus pneumoniae*)

Dear Prof. Yutaka Terao:

On the basis of recommendations from expert reviewers in the field, I have determined that your manuscript requires edits before acceptance. The reviewers found the study to overall be of interest to the field, but noted several points for clarification or where additional support is needed. While you are revising the manuscript, please specifically address the comments copied below from the Reviewers. Reviewer 1's first comment should be addressed experimentally to support the rigor and reproducibility of the study. Keep in mind to all claims must be experimentally supported, so avoid speculation or interpretation beyond that which is specifically demonstrated in an experiment. For example, in vivo effect or specific action on lipoproteins should be either experimentally supported along the Reviewer recommendations, or the referring text revised. Please also carefully check grammar and other writing throughout, referring to journal guidelines with regard to all formatting requirements.

Link Not Available

Sincerely,

Christopher LaRock

Journals Department
Reviewer comments:

Reviewer #1 (Comments for the Author):

In the manuscript entitled, "Macrolides decrease the pro-inflammatory activity of macrolide-resistant *Streptococcus pneumoniae*", Domon et al demonstrate that macrolide treatment decreases release of TLR2 and NOD2 agonists, reducing the pro-inflammatory activity of MRSP. They further show that macrolide treatment of MRSP in vitro downregulates expression of some genes responsible for peptidoglycan synthesis, as well as LTA and lipoprotein synthesis - they suggest that this may be the mechanism resulting in reduced pro-inflammatory activity of macrolide-treated MRSP strains. While the study is interesting, it would benefit from the addition of some controls, and conclusively establishing the contribution of macrolides in reducing lipoprotein levels in the cell.

Major Comments -

1. Control - Macrolides are known to have an immunomodulatory effect on host cells. In this study, the authors are arguing that macrolides impact pneumococcal components as well rather than exerting immunosuppressive effects themselves and have included certain relevant controls in Fig 1B. It would also be beneficial to demonstrate that macrolide treatment does not directly impact NF- κ B activation in the second cell line they have used in Figs 2 and 5.
2. Authors speculate that modest reductions in LTA or lipoprotein-synthesizing enzyme levels in the cell upon macrolide treatment are responsible for reduced activation of TLR2 signaling (Fig 7). However, they were unable to show any difference in LTA levels upon macrolide treatment due to technical constraints (Lines 209-211). The manuscript does not include any data showing that macrolides indeed lead to reduced lipoprotein levels in the cell. Can authors test that? Alternatively, it might be worth including *lgt* or *lsp*-deficient mutant and test if macrolide treatment of these strains has any impact on TLR2 activation.

Minor Comments -

1. Fig 1: X-axis labels unclear in Fig 1A. What is "AZM" and "ERY" - are these macrophages treated with the antibiotics directly? Please include in the legend. Further, please correct X-axis labels in Fig 1C so they are consistent throughout the manuscript (AM and EM).
2. Please specify TNF-alpha as opposed TNF throughout the manuscript.
3. Line 74-76: Unclear - please rephrase.
4. Typos: delete "primarily" (line 62), 32% (line 67).

Reviewer #2 (Comments for the Author):

The authors evaluated the pro-inflammatory activity of macrolide-resistant *S. pneumoniae*. The work was interesting and nicely done. Please address the following questions and comments.

Major comments

1. While the paradox of in vitro resistance and in vivo efficacy has been reported and studied, lines 26-27 "macrolides are clinically effective for treating these diseases, regardless of the susceptibility of the causative pneumococci to macrolides" is a bold statement and likely over-generalized. The anti-inflammatory action of macrolide does not equate "treating these diseases". Clinically there are infections caused by strains of *S. pneumoniae* with macrolide resistance that do respond to macrolide treatment. Also, in some studies, e.g. PMID 25807239, only a very small fraction of the patients with MRSP received macrolide monotherapy. It is sometimes difficult to evaluate the effect of macrolide resistance on patient outcome.
2. What is sub-MIC (line 332)? How was the concentration of the macrolide used in this study determined?
3. The studies consist of macrophage cell culture in the presence or absence of macrolides. Why did the author suggested "our findings indicate... in MRSP in vivo..." (lines 346-347)? There was no animal study involved. Along that line, the authors proved that macrolide reduced pro-inflammatory activity of MRSP in vitro, but there were not enough evidence to show in vivo efficacy. Please comment.

Minor comments

1. Lines 30-32 and lines 41-42 were redundant.
2. Lines 42-44 were redundant in the abstract.
3. Lines 73-74: The sentence was overgeneralized. A few case reports as reference cannot rule out the efficacy of macrolide on pneumococcal infections. The "assumption" was not accurate.
4. Line 75-76: The use of "in contrast" was not appropriate, given the logic of the two sentences about CAP treatment.
5. Line 85: MRSP pulmonary infections

Staff Comments:

Preparing Revision Guidelines

Please return the manuscript within 60 days; if you cannot complete the modification within this time period, please contact me. If you do not wish to modify the manuscript and prefer to submit it to another journal, please notify me of your decision immediately so that the manuscript may be formally withdrawn from consideration by Microbiology Spectrum.

Response to Reviewers

Reviewer #1

Response to the Reviewer 1:

We are grateful to the reviewer 1 for the critical comments and suggestions that have helped us to improve our paper considerably. According to your comments, we performed some additional experiments and added several figures according to your comments. As indicated in the responses that follow, we have taken all these comments and suggestions into account in the revised version of our manuscript.

<Comment #1>

1. Control - Macrolides are known to have an immunomodulatory effect on host cells. In this study, the authors are arguing that macrolides impact pneumococcal components as well rather than exerting immunosuppressive effects themselves and have included certain relevant controls in Fig 1B. It would also be beneficial to demonstrate that macrolide treatment does not directly impact NF- κ B activation in the second cell line they have used in Figs 2 and 5.

<Response>

Figure 2F-J and lines 142-146 (in the revised version):

According to the reviewer's suggestion, we added Figure 2F-J to assess whether macrolide treatment directly affect NF- κ B activation in HEK-Blue cells. In this regard, we added some sentences in Results section (lines 142-146) as follows: "To assess whether macrolides exerted anti-inflammatory effects on HEK-Blue cells, we mixed the supernatants from macrolide-untreated MRSP cells with AZM or ERY and added them to the cultures. Fig. 2F-J depicts that macrolides did not significantly alter SEAP activity in pneumococcal supernatant-treated HEK-Blue cells."

In Fig 5, we have used ethanol-killed MRSP cells. To prepare the MRSP cell samples, pneumococcal cells were treated with ethanol, followed by a wash in PBS. Thus, the samples do not contain macrolides. We believe that "macrolide-untreated MRSP cells + macrolide" groups are not required in Fig 5. Additionally, as shown in Fig2F-J, macrolide treatment did not directly decrease NF- κ B activation in HEK-Blue cells.

<Comment #2>

2. Authors speculate that modest reductions in LTA or lipoprotein-synthesizing enzyme levels in the cell upon macrolide treatment are responsible for reduced activation of TLR2 signaling (Fig 7). However, they were unable to show any difference in LTA levels upon macrolide treatment due to technical constrains (Lines 209-211). The manuscript does not include any data showing that macrolides indeed lead to reduced lipoprotein levels in the cell. Can authors test that?

Alternatively, it might be worth including *lgt* or *lsp*-deficient mutant and test if macrolide treatment of these strains has any impact on TLR2 activation.

<Response>

Fig 8B, Fig 10D, Fig S2, and lines 232-238, 256-257, and 465-482 (in the revised version):

Thank you for your helpful comment. We performed Triton X-114 phase separation (Ref# 34), which can extract lipoprotein from bacterial cells (Ref# 35, 36), and added Fig 8B and Fig 10D. We also added some sentences in Results section as follows: “Previous studies reported that Triton X-114 phase partitioning extracts lipoproteins from bacterial cells (34, 35), including *S. pneumoniae* (36). We investigated the expression of lipoproteins in macrolide-treated MRSP using Coomassie blue staining of Triton X-114 extracts. Fig. 8B depicts that treatment with both AZM and ERY decreased the intensity of lipoproteins in MRSP. Additionally, a 37-kDa lipoprotein, one of the major bands detected in macrolide-untreated MRSP cells, was almost absent in the Triton X-114 extract from macrolide-treated MRSP.” in lines 232-238, and “Fig. 10D depicts that treatment with both AZM and ERY decreased the intensity of lipoproteins in the MRSP strain KM2412.” in line 256-257. In this regard, Materials and Methods section (lines 465-482) was updated. We also added a data using a commercial LTA ELISA kit in Fig S2.

<Comment #3>

1. Fig 1: X-axis labels unclear in Fig 1A. What is "AZM" and "ERY" - are these macrophages treated with the antibiotics directly? Please include in the legend. Further, please correct X-axis labels in Fig 1C so they are consistent throughout the manuscript (AM and EM).

<Response>

Lines 698-701 and Fig 1C (in the revised version):

To avoid confusion, we corrected the legend as follows: “(A) THP-1-derived macrophages were exposed to 2-fold diluted culture supernatant (Sup) from untreated or macrolide-treated MRSP or RPMI 1640 medium containing 2.5 µg/mL macrolides (AZM or ERY) for 9 h.”

We also corrected X-axis in Fig 1C.

<Comment #4>

2. Please specify TNF-alpha as opposed TNF throughout the manuscript.

<Response>

According to the reviewer’s suggestion, we corrected “TNF” to “TNF-alpha” in the manuscript and Figures.

<Comment #5>

3. Lines 74-76: Unclear - please rephrase.

<Response>

Lines 72-75 (in the revised version):

According to the reviewer's suggestion, we rephrased the sentences as follows: "Although several case reports have highlighted macrolide treatment failure during MRSP infections such as CAP and meningitis (5), azithromycin (AZM) therapy has demonstrated clinical efficacy and bacterial eradication for the treatment of CAP caused by MRSP (6, 7)." in lines 72-75.

<Comment #6>

4. Typos: delete "primarily" (line 62), 32% (line 67).

<Response>

Lines 61 and 66 (in a revised version):

Thank you for pointing out our typos. We deleted "primarily" and corrected "32" to "32%".

Response to the Reviewer 2:

We thank the reviewer 2 for the critical comments and suggestions that have helped us to improve our paper considerably. As indicated in the responses that follow, we have taken all these comments and suggestions into account in the revised version of our paper.

<Major comment #1>

1. While the paradox of *in vitro* resistance and *in vivo* efficacy has been reported and studied, lines 26-27 "macrolides are clinically effective for treating these diseases, regardless of the susceptibility of the causative pneumococci to macrolides" is a bold statement and likely over-generalized. The anti-inflammatory action of macrolide does not equate "treating these diseases". Clinically there are infections caused by strains of *S. pneumoniae* with macrolide resistance that do respond to macrolide treatment. Also, in some studies, *e.g.* PMID 25807239, only a very small fraction of the patients with MRSP received macrolide monotherapy. It is sometimes difficult to evaluate the effect of macrolide resistance on patient outcome.

<Response>

Lines 26-27 (in the revised version):

Thank you for your helpful comments. According to the reviewer's suggestion, we have rephrased the sentence as follows: "macrolides may be clinically effective for treating these diseases, regardless of the susceptibility of the causative pneumococci to macrolides." in line 26-27.

<Major comment #2>

2. What is sub-MIC (line 332)? How was the concentration of the macrolide used in this study

determined?

<Response>

To avoid confusion, we deleted the word “sub-MIC” in line 348 (in the revised version).

It has been reported that mean \pm SD plasma erythromycin levels after oral administration of 500 and 1000 mg erythromycin reaches 3.8 ± 1.4 and 6.5 ± 2.9 $\mu\text{g/ml}$ (Josefsson K, et al. Br J Clin Pharmacol. 13(5): 685-91. 1982.). Also, Wollmer et al demonstrated that the mean extravascular concentrations in lung obtained during the first hour after an intravenous injection of 270 mg erythromycin reach $5.5\text{--}6.6 \pm 2.2$ $\mu\text{g/g}$ (Lancet, 2:1361-4, 1982). Therefore, we decided to add 4-5 $\mu\text{g/ml}$ macrolides into MRSP cultures in this study.

<Major comment #3>

3. The studies consist of macrophage cell culture in the presence or absence of macrolides. Why did the author suggested "our findings indicate... in MRSP *in vivo*..." (lines 346-347)? There was no animal study involved. Along that line, the authors proved that macrolide reduced pro-inflammatory activity of MRSP *in vitro*, but there were not enough evidence to show *in vivo* efficacy. Please comment.

<Response>

Lines 354-365 (in the revised version):

Although animal experiments were performed only in our previous study, we inadvertently gave the impression that we conducted animal experiments in the current study. To avoid confusion, we corrected some sentences as follows: “Our previous study demonstrated that oral administration of macrolides to mice intratracheally infected with MRSP resulted in decreased IL-6 protein levels in bronchoalveolar lavage fluid compared to untreated infected control mice without affecting the bacterial load in the fluid (15). Additionally, Yasuda et al. reported that the administration of roxithromycin, a 14-membered macrolide that does not inhibit PLY production, increased the median survival time and retarded bacteremia in a murine pneumonia model with MRSP (51). Although the immunomodulatory effects of macrolides may partly contribute to their *in vivo* efficacy, these findings suggest that macrolides may also affect the transcription of pro-inflammatory component-related genes in MRSP *in vivo* (15). Collectively, our previous and present findings provide additional explanations for the clinical benefits of macrolides.” in lines 354-365.

<Minor comments #1-3>

1. Lines 30-32 and lines 41-42 were redundant.
2. Lines 42-44 were redundant in the abstract.
3. Lines 73-74: The sentence was overgeneralized. A few case reports as reference cannot rule

out the efficacy of macrolide on pneumococcal infections. The "assumption" was not accurate.

<Response>

According to the reviewer's suggestion, we deleted lines 30-32, 41-42, and 73-74.

On the other hand, since we performed Triton X-114 separation and found that lipoprotein expressions were decreased in macrolide-treated MRSP cells than in untreated MRSP cells, we left lines 42-44 (lines 41-43 in the revised version).

<Minor comment #4>

4. Lines 75-76: The use of "in contrast" was not appropriate, given the logic of the two sentences about CAP treatment.

<Response>

Line 72-75 (in a revised version):

According to the reviewer's suggestion, we rephrased the sentences as follows: "Although several case reports have highlighted macrolide treatment failure during MRSP infections such as CAP and meningitis (5), azithromycin (AZM) therapy has demonstrated clinical efficacy and bacterial eradication for the treatment of CAP caused by MRSP (6, 7)." in line 72-75.

<Minor comment #5>

5. Line 85: MRSP pulmonary infections

<Response>

Line 84 (in the revised version):

We corrected "MRSP infection" to "MRSP pulmonary infections" in line 84.

April 25, 2023

Prof. Yutaka Terao
Niigata University Graduate School of Medical and Dental Sciences
Division of Microbiology and Infectious Diseases
5274 Gakkocho 2-ban-cho, Chuo-ku
Niigata 951-8514
Japan

Re: Spectrum00148-23R1 (Macrolides decrease the pro-inflammatory activity of macrolide-resistant *Streptococcus pneumoniae*)

Dear Prof. Yutaka Terao:

Your manuscript has been accepted, and I am forwarding it to the ASM Journals Department for publication. You will be notified when your proofs are ready to be viewed.

Sincerely,

Christopher LaRock
Editor, Microbiology Spectrum
